# An evolutionary approach to predict the orientation of CRISPR arrays

Axel Fehrenbach[1,2], Alexander Mitrofanov[3], Omer S. Alkhnbashi[4,5], Rolf Backofen[3,6], Franz Baumdicker[1,2]*

1 Cluster of Excellence "Controlling Microbes to Fight Infections", Mathematical and Computational Population Genetics, University of Tübingen, Tübingen, Germany, 2 Institute for Bioinformatics and Medical Informatics (IBMI), University of Tübingen, Tübingen, Germany, 3 Bioinformatics group, Department of Computer Science, University of Freiburg, Freiburg, Germany, 4 Center for Applied and Translational Genomics (CATG), Mohammed Bin Rashid University of Medicine and Health Sciences (MBRU), Dubai Healthcare City, Dubai, United Arab Emirates, 5 College of Medicine, Mohammed Bin Rashid University of Medicine and Health Sciences (MBRU), Dubai Healthcare City, Dubai, United Arab Emirates, 6 Signalling Research Centres BIOSS and CIBSS, University of Freiburg, Freiburg, Germany

* franz.baumdicker@uni-tuebingen.de

## Abstract

CRISPR-Cas is a defense system of bacteria and archaea against phages. Parts of the foreign DNA, called spacers, are incorporated into the CRISPR array which constitutes the immune memory. The orientation of CRISPR arrays is crucial for analyzing and understanding the functionality of CRISPR systems and their targets. Several methods have been developed to identify the orientation of a CRISPR array. To predict the orientation, different methods use different features such as the repeat sequences between the spacers, the location of the leader sequence, the Cas genes, or PAMs. However, those features are often not sufficient to predict the orientation with certainty, or different methods disagree.

Remarkably, almost all CRISPR systems have been found to insert spacers in a polarized manner at the leader end of the array. We introduce *CRISPR-evOr*, a method that leverages the resulting patterns to predict the acquisition orientation for (a group of) CRISPR arrays by reconstructing and comparing the likelihood of their evolutionary history with respect to both possible acquisition orientations. The new method is independent of Cas type, leader existence and location, and transcription orientation.

*CRISPR-evOr* is thus particularly useful for arrays that other CRISPR orientation tools cannot predict confidently and to verify or resolve conflicting predictions from existing tools. *CRISPR-evOr* currently confidently predicts the orientation of 28.3% of the arrays in the considered subset of CRISPRCasdb, which other tools like CRISPRDirection and CRISPRstrand cannot reliably orient. As our tool leverages evolutionary information we expect this percentage to grow in the future when more closely related arrays will be available. Additionally, *CRISPR-evOr* provides confident

**Data availability statement:** The tool CRISPR-evOr is provided within the SpacerPlacer framework at https://github.com/fbaumdicker/SpacerPlacer. All relevant data are within the manuscript and its Supporting information files.

**Funding:** AF and FB are funded by the Deutsche Forschungsgemeinschaft (DFG, German Research Foundation) under the Priority Program - SPP 2141 - Project number Ba-5529/1-1. FB is funded by the Deutsche Forschungsgemeinschaft (DFG, German Research Foundation) under Germany's Excellence Strategy – EXC number 2064/1 – Project number 390727645, and EXC 2124 – Project number 390838134. RB is funded by the German Research Foundation (DFG) [BA 2168/23-1 and BA 2168/23-2 SPP 2141]; Much more than Defence: the Multiple Functions and Facets of CRISPR–Cas; Baden-Wuerttemberg Ministry of Science, Research and Art; University of Freiburg. RB is funded by the BMBF-funded de.NBI Cloud within the German Network for Bioinformatics Infrastructure (de.NBI) (031A532B, 031A533A, 031A533B, 031A534A, 031A535A, 031A537A, 031A537B, 031A537C, 031A537D, 031A538A). The funders had no role in study design, data collection and analysis, decision to publish, or preparation of the manuscript.

**Competing interests:** The authors have declared that no competing interests exist.

decisions for rare subtypes of CRISPR arrays, where knowledge about repeats and leaders and their orientation is limited.

---

## Author summary

Some bacteria and archaea possess a CRISPR-Cas defense system, which protects them against phages and mobile genetic elements. This system adapts to new threats by incorporating small fragments of their DNA, so called spacers, in a CRISPR array. Remarkably, the acquisition of new spacers is polarized at one end of this array. To understand how this immune system functions, it is essential to know the orientation of these arrays. Many existing tools try to determine orientation using genetic markers, but these methods are often unreliable or disagree with one another.

In our work, we developed a new method that predicts the end at which new spacers are inserted, by considering the evolutionary history of a group of related CRISPR arrays. Unlike other tools, our approach is less reliant on specific genetic markers and can be applied broadly across many types of CRISPR systems. We show that it can confidently determine the orientation of a large number of arrays that other methods cannot resolve. This provides a new way to predict the array orientation, which is particularly useful for rare CRISPR types. Our evolutionary approach will become even more powerful as more genetic data becomes available.

## Introduction

CRISPR-Cas systems are powerful defense systems of bacteria and archaea and provide adaptive immunity against foreign elements like phages and plasmids [1,2]. One component of the system are the so called CRISPR associated (Cas) genes that allow the bacterium to produce protein complexes that are able to interact with mobile genetic elements. Depending on the composition of Cas genes CRISPR-Cas systems are classified into a multitude of classes and types by the Cas genes found in the genome and the presence of CRISPR arrays [3,4]. The other component is the CRISPR array and composed of repeats which are interspaced by small DNA snippets called spacers. These spacers are expressed as crRNA which fit into Cas protein complexes to target specific foreign elements and destroy or block them [1,2].

The systems adapt to new threats by acquiring new spacers from snippets of invader sequences which are then inserted in the CRISPR array with a new repeat unit [2,5]. Notably, new spacers are almost always inserted at the so called leader end [5,6]. The leader is a non-coding sequence of variable length (up to a few hundred bp) with limited sequence conservation that is located, in general, at the 5' end of the array. The leader is responsible for regulating the acquisition and transcription of spacers [7–10].

CRISPR arrays evolve over time by the acquisition and deletion of spacers. Moreover, due to the polarized insertion process, spacer arrays constrain the evolutionary history as they are ordered by the time of their acquisition.

## Importance of orientation for CRISPR systems

DNA and RNA are both transcribed and synthesized in the 5'-3' direction. Thus, the orientation of CRISPR arrays carries importance in a multitude of facets, including the processing of spacers into crRNA, the interference and adaptation mechanisms and thus the understanding of different Cas types. In particular, it is essential to identify leader regions, to identify transcription initiation and termination regions, to determine the orientation of protospacers and to identify PAM sequences [9,11,12]. Furthermore, inserting foreign arrays into new organisms in the wrong orientation may cause them to be mischaracterized as non-functional in experiments.

The orientation is also of crucial importance when investigating the ecology and evolution of CRISPR arrays. The acquisition process of most CRISPR arrays was found to be polarized [1,2]. Thus, if the orientation prediction is wrong the whole ordering would be interpreted wrongly and evolutionary inference is likely to produce false results [13].

## Orientation concepts and tools for CRISPR-Cas systems

In the following, we introduce the concepts and the corresponding tools to predict the CRISPR array orientation. Tools can provide differing predictions for the orientation of CRISPR arrays due to different orientation concepts, but also between the predictions of tools using the same concept. We give a short overview of the different concepts in Table 1.

The *transcriptome* can be used to infer the orientation of arrays [21], however, this information is typically not available. We thus focus on other available concepts and tools here.

In most known CRISPR arrays found in nature, there are Cas genes in close proximity. Thus, Milicevic et al. suggest to predict the CRISPR orientation based on the "Cas-orientation". The Cas-orientation is determined by the transcription orientation of the Cas genes close to a respective CRISPR array (< 1000 bp distance) [14].

Furthermore, CRISPR arrays are highly characterized by their *leader*, which is found at only one end of the array. Typically, the leader is found at the 5' end. Thus, one can determine the "leader-orientation" by determining the location of the leader.

A closely related orientation concept is based on the *repeats*. The "repeat-orientation" is characterized by mutation patterns along the array. Due to the molecular functionality of the (polarized!) insertion and deletion process of spacers, repeats degrade in 5'-to-3' direction by accumulating mutations [18], moreover, the mutations are concentrated at the ends of repeats [15].

**Table 1**. **Orientation concepts overview.** We show all orientation concepts in use for CRISPR arrays that are referenced within this work. We give them labels to simplify referencing the different orientation concepts. We also provide references for the concepts and tools that rely on the respective orientation concept.

| Orientation concept | Description and Tools/Papers using the concept |
|---|---|
| acquisition | Find the end of the array where new spacers are inserted by analyzing the evolution of groups of CRISPR arrays. **Tool:** *CRISPR-evOr* (new) |
| Cas | Determine the transcription orientation of Cas genes close to the array and identify their orientation with the array orientation. **Reference:** Milicevic et al. [14] |
| PAM | Find PAMs of CRISPR systems and locate the PAMs on matched targets of spacers to determine the orientation. **Reference:** Vink et al. [11] |
| repeat | Characterize the repeats according to available biological knowledge and their relation to transcription orientation. In particular, analyze the mutation pattern of the repeats along the length of the array. **Tools:** CRISPRstrand [15], CRISPRidentify [16], CRISPRmap [17] |
| leader + repeat | Use empirically measured characteristics to analyze the repeats and locate the leader sequence by analyzing the regions adjacent to the CRISPR array. These characteristics are weighted and combined to generate a prediction. **Tools:** CRISPRDirection [18], CRISPRCasFinder [19], CRISPRDetect [20] |
| transcriptome | Predict orientation by employing transcriptome sequencing to find the transcription orientation of arrays. **Tool:** TOP [21] |

 

Both the leader- and repeat-orientation are closely related concepts and one of the most commonly used prediction tools for CRISPR arrays, CRISPRCasFinder [19], relies on both leader- and repeat based orientation methods. More precisely, CRISPRCasFinder employs a combination of leader- and repeat-orientation [19] and incorporates "CRISPRDirection" [18]. The orientation is predicted by, firstly, trying to find the leader by searching for relative AT richness at both ends of the array and comparing the distance to the closest coding genes at both ends. Secondly, CRISPRDirection considers specific sequence motifs and mutations, the RNA secondary structure, AT content in the repeats, and degeneracy in the 3' end of the array [19,20].

"CRISPRidentify" [16] is another commonly used tool to find CRISPR systems that relies on repeat based orientation by incorporating "CRISPRstrand" [15]. CRISPRidentify does not use any additional information about the leader for the orientation prediction, but CRISPRleader within CRISPRidentify can annotate the leader [9] by using the repeat-orientation predicted by CRISPRstrand. CRISPRstrand uses domain expert knowledge about repeats to identify important features. It was observed that it is useful to partition the consensus repeat into different blocks as the mutation pattern along the consensus repeat strongly diverges. CRISPRstrand encodes the consensus repeats as graphs, which additionally, contain information about mutations and their position in the repeat. Then they train a graph kernel model on the encoded graphs of a curated dataset [15].

Vink et al. identified the orientation of many CRISPR arrays by identifying the respective protospacers of the spacers in the array and *locating the protospacer adjacent motifs* (PAM) [11]. PAMs are (generally) 3 nucleotide long sequences at one side of the protospacer. They are used by the CRISPR-Cas system to correctly distinguish between spacers and protospacers to avoid autoimmunity. These sequences are quite specific to the systems Cas-(sub)type.

While all of these concepts are viable ways to predict the orientation, they not necessarily agree and do not always provide a reliable basis for a confident prediction. For example, for some CRISPR arrays Cas genes will be reversed or no Cas genes are located closely to the array, a leader sequence does not exist [9], no protospacers can be found, or the CRISPR arrays are too short to analyze the repeats. In particular, type II-C systems have been shown to be peculiar in multiple ways: they acquire spacers at the 3' end (opposite to the expected leader end) [22–24], carry promoters in the repeats [22,24,25] and show peculiar repeat mutation patterns [26] all of which could possibly interfere with accurate orientation inference. Consequently, novel concepts and methods for array orientation are necessary to expand the range of confidently orientable arrays.

## Acquisition-orientation as a holistic evolutionary approach to predict array orientation

In contrast to the features used by previous approaches of orientation prediction, one (almost) universal property of CRISPR array evolution is the polarized acquisition of new spacers into the array. In most cases, acquisition is facilitated at the leader end, i.e. the 5' end [5,6]. These polarized insertions lead to highly distinctive evolutionary patterns in the reconstructed ancestral history of groups of closely related CRISPR arrays.

We propose *CRISPR-evOr* a method that identifies what we call the "acquisition-orientation", the most likely end of acquisition. This orientation concept is closely related to the other orientation concepts, in particular, the leader- and repeat-orientation. However, instead of relying on markers in single CRISPR arrays or genomes, *CRISPR-evOr* uses a holistic approach that relies on reconstructions of the evolution of a group of CRISPR arrays. For the reconstructions, we rely on our recently introduced tool *SpacerPlacer* [13]. *SpacerPlacer* reconstructs the ancestral history of spacer acquisitions and deletions of a group of arrays (with spacer overlap) along a phylogenetic tree by using maximum likelihood methods and leveraging the spacer insertion order (inferred from the arrays).

*CRISPR-evOr* relies on a simple heuristic. We reconstruct the ancestral history with *SpacerPlacer* [13] twice. First, the evolution is reconstructed for the arrays in the given orientation. Then we reverse the CRISPR array spacer sequence and compute a second reconstruction. Subsequently, *CRISPR-evOr* compares the likelihoods of each reconstruction provided

by *SpacerPlacer* to predict the most likely acquisition-orientation. In this way, we acquire an orientation prediction by consensus decision of all CRISPR arrays in the group. The workflow of *CRISPR-evOr* is portrayed in Fig 1. Fig 2 shows

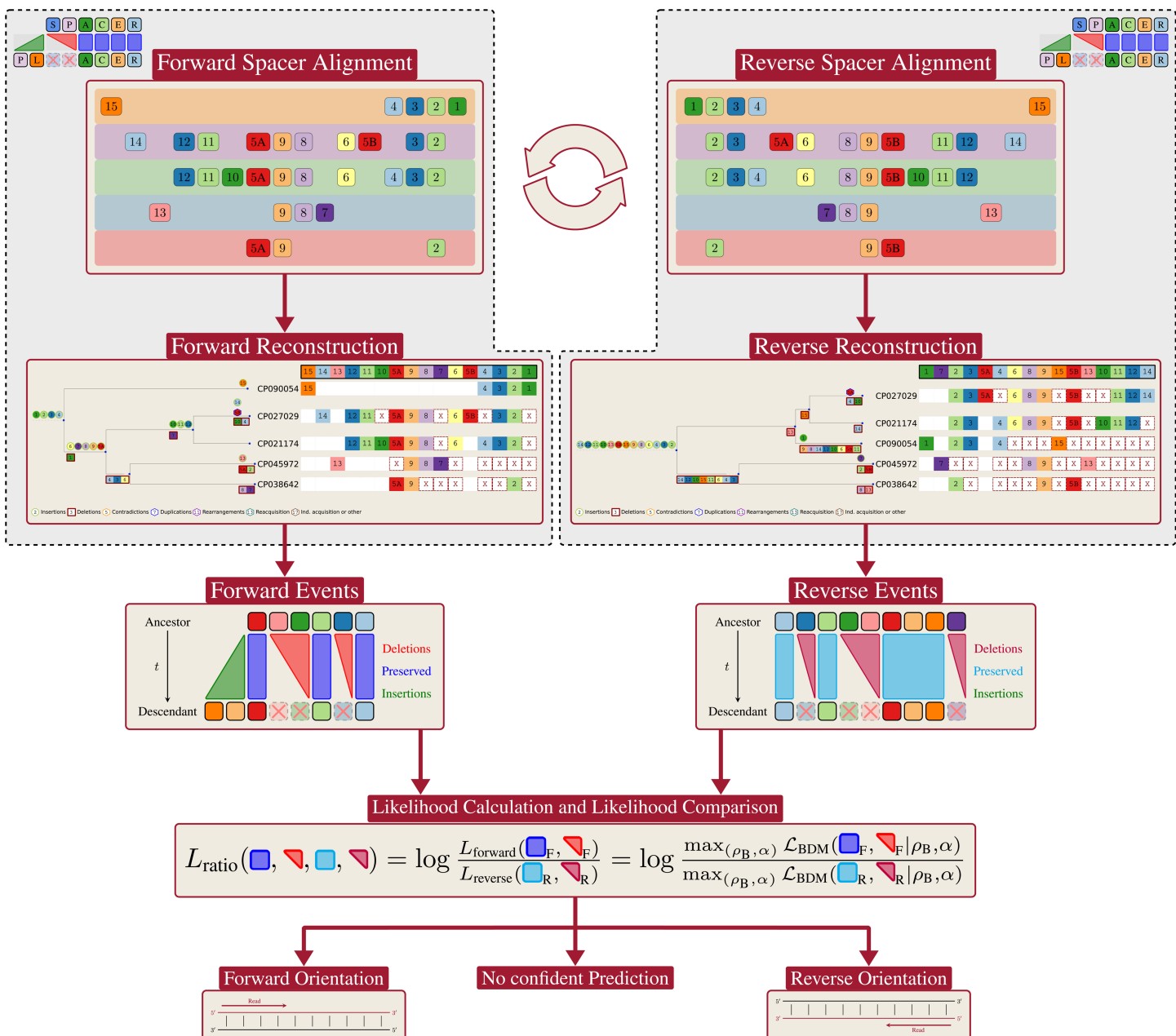

**Fig 1**. *CRISPR-evOr* **workflow.** We show the steps taken by *CRISPR-evOr* to arrive at a prediction for a given group of spacer alignments. The left and right column show the same steps of reconstructing the ancestral history of a spacer alignment with *SpacerPlacer*. They differ only by their orientation, i.e. the spacer alignments are turned around compared to each other. Note, that *SpacerPlacer* is heavily reliant on the partial spacer insertion order (PSIO) and thus, this difference can have major impact on the reconstruction. Furthermore, in this case here, no tree was provided to *SpacerPlacer* and thus, *SpacerPlacer* estimates a different tree based on the spacer arrays for each orientation. After both reconstructions are performed, *SpacerPlacer* obtains maximum likelihoods of the respective reconstructions based on the reconstructed events. Then by comparing the ratio of both likelihoods, *CRISPR-evOr* decides for one of the orientations or, if the likelihoods are too close, determines that no confident prediction can be made.

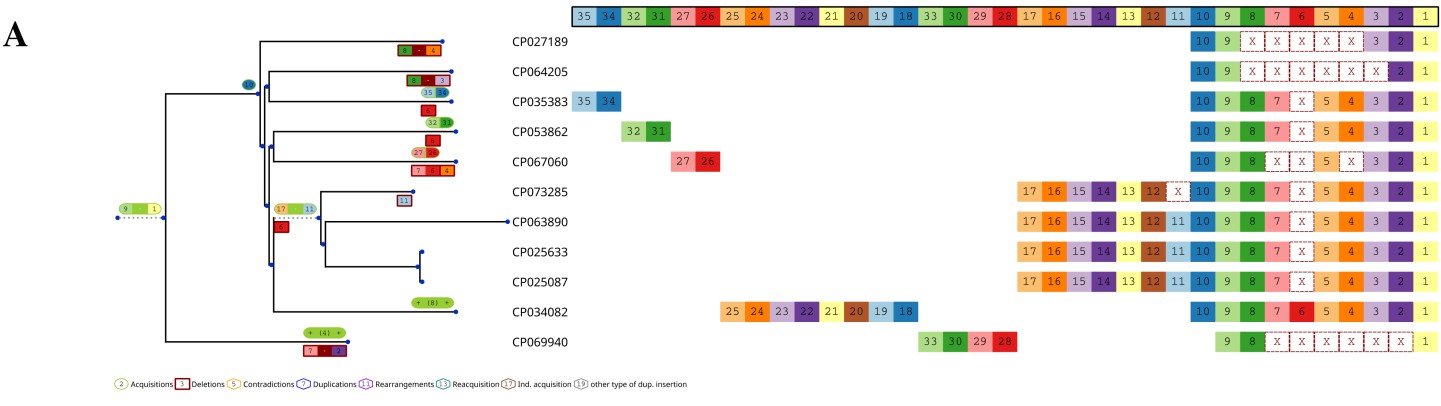

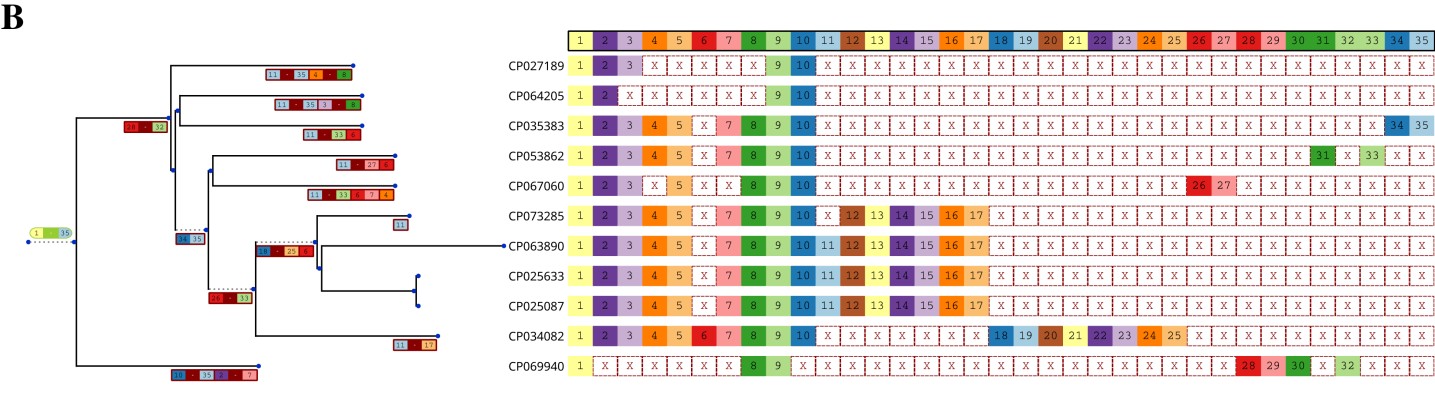

**Fig 2**. **Example of a group of arrays with high confidence orientation prediction. A** and **B** show reconstructions of the same alignment of spacer arrays. B has the orientation provided by CRISPRCasdb, while A is simply the reconstruction of the same arrays, but arranged in reverse order. Note, that in B all spacers are reconstructed to be inserted (green) at the root and thus the exceptional amount of deletion events (red) along the tree (also shown in red in the spacer alignment). The insertions are produced through the first spacer 1 (light yellow) which is present at the root, as it is present in all arrays, and "pulls up" all other spacer insertions through the spacer insertion order. In contrast, in A the insertions are distributed more evenly along the tree and thus the reconstruction avoids a lot of deletions. Comparing both reconstructions clearly indicates that the orientation of A is more likely under the assumption of polarized insertions. To make the figures more concise, acquisition and deletion events in the tree are consolidated, e.g. 9 - 1 in green/red indicates that spacers 9 through 1 were acquired/deleted at this branch. At the leafs "+ (x) +" indicates that x spacers were acquired. They are the leftmost spacers in the alignment for the respective leafs (which are not found in any other array). The shown example is a group of *Klebsiella pneumoniae* (Cas type I-E) from CRISPRCasdb with small adjustments for readability.

an example with a high confidence prediction where, clearly, the acquisition-orientation shown in A can be favored over the reverse as shown in B, simply due to the much smaller number of reconstructed evolutionary events.

We apply *CRISPR-evOr* on a large database of CRISPR arrays extracted from CRISPRCasdb [27] and find that *CRISPR-evOr* confidently predicts orientations for arrays where the aforementioned orientation methods have low confidence. Quantitatively, *CRISPR-evOr* can predict the orientation of 28.3% of all arrays where CRISPRDirection and CRISPRstrand have low confidence. Thus, our tool increases the number of arrays where orientations can be reliably estimated.

Furthermore, on data where high confidence orientation predictions are already available, *CRISPR-evOr* helps to either reinforce the predictions, or resolve contradicting predictions of other tools, and thereby uncovers discrepancies among different orientation concepts dependent on the Cas type. We find that among the array subsets where multiple tools are confident, predictions by *CRISPR-evOr* almost always agree with other tools for two of the most abundant Cas types

(I-E and I-F) in CRISPRCasdb. In particular for types I-B and II-A, *CRISPR-evOr* agrees more often with CRISPRstrand than CRISPRDirection. The differences indicate that CRISPRDirection is well-attuned only for arrays of some Cas types. Moreover, *CRISPR-evOr*, PAM-orientation, CRISPRDirection, and CRISPRstrand often disagree with each other for type II-C, which hints at differences between acquisition-orientation and leader-, repeat- and PAM-orientation for type II-C.

## Methods

### Database and available orientation predictions

We rely on the CRISPRCasdb dataset collected by us in [13]. It is based on high-confidence arrays extracted from CRISPRCasdb [27] in July 2022. For each array the orientation prediction by CRISPRDirection was extracted from CRISPRCasdb and CRISPRstrand was applied to obtain the respective orientation predictions. These arrays were grouped successively by consensus repeat, spacer overlap, genus and Cas type to yield a large comprehensive dataset. For the grouping by spacer overlap, spacers were clustered as described in [13]. Then groups were removed if they did not share any spacer overlap between arrays or if their arrays were completely identical. Thus, each group contains only arrays of the same consensus repeat, genus, Cas type and overlapping spacer content. Examples of such groups are shown in Fig 2 and Figs B, C in S1 Material. The dataset contains a total of 518 groups with 5934 arrays. A breakdown of the Cas types in the dataset is shown in Fig A in S1 Material. Fig 4 shows the Cas type distribution for arrays with high confidence orientation predictions by *CRISPR-evOr*, CRISPRstrand and CRISPRDirection in this dataset.

For more details about the preparation of the groups and the dataset in general, see Methods and Supplementary Note S1A in [13]. Note that in [13] we investigated and estimated parameters of the deletion process on a subset (334 groups, 4565 arrays with at least one deletion in the reconstruction) of the larger dataset used here.

Additionally, we used an updated version of the dataset collected by Vink et al., who extracted their dataset from CRISPRCasdb in February 2020 [11]. The updated version uses the same dataset of spacers, but contains more confident orientation predictions. The old version is available in their supplementary material. The database contains 4266 bacterial and archaeal genomes and a total of 221,591 spacers and their respective consensus repeat sequences. They subsequently matched these spacers against the NCBI nucleotide database and large metagenomic databases to obtain target matches. They then investigated the flanks of the matches to find the associated PAM sequences and predict the orientation of spacers based on the location of the respective PAM at the matching target's flanks.

We matched the repeats and respective spacers against our dataset to allow us to compare the orientation predictions. Thus, the dataset used for the evaluation of PAM-orientation versus *CRISPR-evOr* is the intersection between the dataset from [13] and Vink et al.'s dataset, both of which are part subsets of CRISPRCasdb. Furthermore, only 38.9% (2309) of the arrays and 77.2% (400) of the groups in the CRISPRCasdb dataset have PAM-orientation predictions. The Cas type diversity of this dataset is shown in Fig 3 and can be observed in Figs E, J, N, O in S1 Material.

### Ancestral reconstruction with *SpacerPlacer*

The backbone of *CRISPR-evOr* is the ancestral reconstruction algorithm for groups of closely related CRISPR spacer arrays *SpacerPlacer* [13]. To prepare the spacer data, *SpacerPlacer* labels each unique spacer found in the dataset with a number and aligns the labeled spacer arrays with MAFFT [28]. This results in an Multiple Spacer Array Alignment (MSAA) where each column corresponds to a unique spacer and each row to a CRISPR array. Assuming a direction of acquisition yields a clear timeline of acquisition for each array, where the spacer near the acquisition end is the youngest spacer while the spacers
further removed from the acquisition end are older spacers. *SpacerPlacer* constructs a Partial Spacer Insertion Order (PSIO) from all CRISPR arrays by combining all individual insertion orders together.

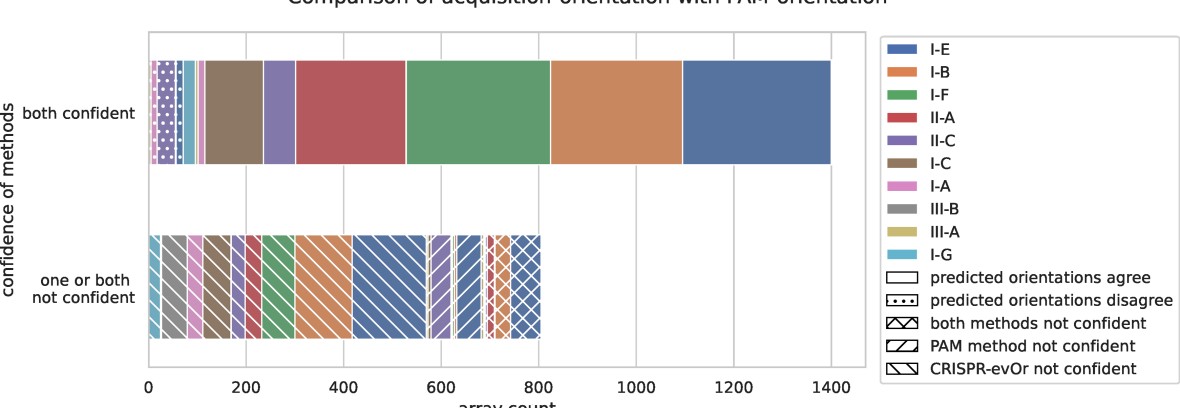

Fig 3. **Comparison between acquisition- and PAM-orientation.** We compare the predictions between the PAM-orientation and *CRISPR-evOr* broken down according to Cas type. Only subtypes with at least 5 groups with more than 50 arrays are shown. While there are many arrays were both methods can not make confident decisions, in cases where they are confident, they almost always agree, but type II-C deviates from this behavior.

The reconstruction proceeds in two steps. In the first step *SpacerPlacer* computes a guide reconstruction based on a straightforward model allowing for independent deletions and insertions at each spacer site. The reconstruction is performed with a joint maximum likelihood approach proposed by Pupko et al. [29]. Naturally, as insertions can occur within the array, the reconstruction will not necessarily follow the PSIO, i.e. spacers can be inserted behind existing spacers.

In the second step *SpacerPlacer* refines the guide reconstruction by enforcing the pre-computed PSIO. The PSIO is enforced by finding contradictions, i.e. insertion events that are behind a leading spacer, along the tree. Each of these contradicting spacers is then fixed to be present at its respective parent node. Then, recomputing the reconstruction with these fixed states yields a new reconstruction, where some of the contradictions might be resolved. This process is repeated until all contradictions are resolved and results in a reconstruction that respects the PSIO. This process always finds a reconstruction without contradictions, where, in the worst case, most or all contradicting spacers end up being acquired at the root.

After both steps, we obtain an ancestral reconstruction that can be analyzed and used to estimate deletion model parameters. Furthermore, the reconstruction is visualized by rendering the respective tree with reconstructed acquisition and deletion events and the MSAA. For an example see Fig 2.

## Computation of likelihoods with *SpacerPlacer*

*SpacerPlacer* is able to compute the likelihoods of reconstructions according to their proposed block deletion model (BDM), which they found to be a suitable model for spacer deletions in statistical analysis on the CRISPRCasdb dataset [13]. We use the simplified likelihood function and default parameters computed on the CRISPRCasdb dataset, as is recommended [13].

## Tree estimation

We estimated trees based on the single-copy core genes of the corresponding sample for each group as is stated in [13]. We used Bakta to annotate genomes [30] and then the pangenome analysis tool panX [31] and IQ-TREE [32] to reconstruct trees. We used these core genome trees for all experiments and figures unless otherwise stated.

If no tree is provided by the user, *CRISPR-evOr* uses the provided functionality of *SpacerPlacer* to estimate trees based on the CRISPR arrays. In that case, *CRISPR-evOr* estimates the tree for each orientation , i.e. reconstruction run, separately since the trees themselves depend on the acquisition orientation and might differ substantially.

*SpacerPlacer* estimates trees using neighbor joining (NJ) or unweighted pair group method with arithmetic mean (UPGMA) with a simple distance function, which estimates the evolutionary distances between a pair of arrays assuming a polarized insertion model with blockwise deletions and suitable empirically estimated rates.

## Simulations

Simulations were performed according to the methods described in [13]. First, a spacer array with length distributed according to the (rough approximation) of the stationary distribution of the BDM is sampled. Subsequently, starting at the root of a given tree, the array is evolved along the tree branches by sampling acquisition and deletion events. The acquisitions are polarized and the deletion events follow the BDM.

To evaluate the performance of *CRISPR-evOr* we simulated two datasets: Dataset A: Here, phylogenetic trees were sampled according to the coalescent and we simulated datasets with reasonable parameter values as described in [13]. Dataset B: For a more realistic setting, we used the core-genome based trees estimated in [13] for the CRISPRCasdb dataset (see tree estimation) and simulated datasets using their respective estimated evolutionary parameters. Note that the parameters estimated by [13] were estimated on groups with at least one observed deletion events, and thus all groups in the simulations have non-zero acquisition and deletion rates.

More information and precise numbers can be found in Note A in S1 Material. Additionally, information about the simulation process can be found in the respective section and Supplementary Note S1 of [13].

## Simulation-based benchmarking

In simulations along coalescent trees and moderate spacer acquisition and deletion parameters (dataset A) we found that *CRISPR-evOr* can be completely relied upon with 100% of the decision being correct for decision threshold $c = 0$. Furthermore, almost all simulations exceed any reasonable choice of threshold $c$.

To investigate the reliability of *CRISPR-evOr* in a more realistic setting (dataset B), we tested *CRISPR-evOr* providing the original trees (inferred from core genes in [13] and upon which we simulated the dataset). However, since reliable trees are not always available, we also tested the performance based on trees inferred by *SpacerPlacer* based on the spacer arrays themselves.

When using the original tree and decision threshold $c = 0$, we found that 97.8% of the predictions were correct. Of the remaining 2.2%, only 0.2% give the wrong orientation prediction and 2% likelihood ratio exactly 0, i.e. no or symmetric events. When using the trees estimated based on the spacer arrays, we found that 95.5% of the predictions were correct and 1.5% have likelihood ratio exactly 0, similar to before. As the estimated trees based on the arrays can be of low resolution and diverge from the true tree, especially for groups with few observed evolutionary events, it comes at no surprise that *CRISPR-evOr* predicted the wrong orientation for 3.0% of the groups. However, imprecise tree inference does not change the underlying observed events and thus does not change results by a huge margin.

Naturally, when a decision threshold $c > 0$ is applied, the percentage of correct and wrong predictions decreases and some of the orientation predictions are labeled as uncertain instead. We report the results on the above simulation datasets with a conservative choice of $c = 5$ in Note B in S1 Material.

## Confidence threshold $c$

We manually compared the forward and reverse reconstructions for CRISPRCasdb groups with low $|L_{ratio}|$ to find a conservative candidate for the confidence threshold with $c = 5$.

For additional evidence that the threshold is suitable, we considered the difference in the number of deleted spacers between forward and reverse reconstructions for each group in the CRISPRCasdb dataset. In general, we find that the number of differences increases rapidly with increasing $|L_{ratio}|$ (Fig D in S1 Material).

For $|L_{ratio}| = 0$, we typically do not have differences in the forward and backward reconstructions (see Fig B in S1 Material). For $0 < |L_{ratio}| \leq 5$ the average difference in the number of spacer deletions between forward and backward is 5.5 (median 2). *CRISPR-evOr* still provides an estimate for the correct orientation in these cases. However, since the number of underlying deletion events is low, we believe that these results should be treated with caution. For $|L_{ratio}| > 5$, the average number of differences is 140.3 (median 42) with almost all groups exceeding at least 5. This is a substantial event count considering that arrays in the dataset have a median length of approximately 15 spacers. Combining both investigations leads us to believe that a conservative threshold of $c = 5$ allows a confident prediction of orientations.

## Results

### Orientation algorithm

We now describe the process with which *CRISPR-evOr* predicts the acquisition-orientation. Previous methods of estimating the strand orientation mostly rely on properties of the repeats and neighboring DNA sequence within a *single* genome. We, however, use spacer arrays of a *collection* of samples and leverage the polarized spacer acquisition process, to estimate the orientation. For a reliable acquisition-orientation prediction, overlap in spacers between samples in this group and enough spacer diversity are required. Before starting the reconstruction and prediction process, we identify and relabel unique spacers within the group with integers, as is done by *SpacerPlacer* [13].

The tool *CRISPR-evOr* is provided within the *SpacerPlacer* framework at https://github.com/fbaumdicker/SpacerPlacer. We provide orientation predictions of all tools, accession numbers and some useful information for all groups considered here in S1 Table (csv). The column headers of S1 Table are described in Note E in S1 Material.

The input into the algorithm is a group of spacer arrays. We show examples for such groups in the github repository and in Figs 1, 2 and Figs B, C in S1 Material. *SpacerPlacer* supports the input of array groups as fasta files of either the array DNA sequences of the spacers (repeats are removed) or arrays with already fixed spacer ids. The spacer DNA sequences can be extracted from CRISPRCasdb and CRISPRCasFinder or CRISPRidentify output. Scripts for the conversion and pre-processing for CRISPRCasFinder output are provided as options of *SpacerPlacer*.

If no tree was provided, the reconstruction is preceded by the estimation of a suitable phylogenetic tree based on the CRISPR arrays as described in [13].

Afterwards, *CRISPR-evOr* follows a three step strategy. First, we reconstruct the ancestral states of the provided spacer data, in the orientation it is provided in, with *SpacerPlacer*. The computed maximum likelihood $L_{forward}$ is saved for later use. In a second step, we *reverse* the spacer integer sequences (not the DNA sequence) of each sample. This is equivalent to reversing the orientation of *all* arrays in the dataset. We then estimate the ancestral reconstruction with *SpacerPlacer* in exactly the same way as in the first step. Again, we save the computed maximum likelihood $L_{reverse}$. Note that if the tree was estimated based on the pairwise distance approach from *SpacerPlacer*, the tree estimation is repeated as well, since the distances are highly dependent on the spacer order.

In the third step, we compare the maximum likelihoods computed for each orientation by taking the ratio of both likelihoods to arrive at an orientation prediction.

**Likelihood comparison and measuring confidence.** Since we use the same parameters for the reconstruction and either use a provided tree (which is independent of the spacer sequences) or, if no tree is provided, estimate the trees for each direction individually, we can compare the likelihoods in a meaningful likelihood ratio

$$L_{ratio} := \log L_{forward} - \log L_{reverse}.$$

To quantify the certainty of *CRISPR-evOr* we introduce a confidence threshold $c \geq 0$. However, since the likelihood is non-canonical, the asymptotic distribution of the likelihood ratio is unknown. Therefore, we rely on empirical results in simulations and on available real data to determine a suitable confidence threshold $c$ to make reliable decisions; see methods.

Computing the $L_{\text{ratio}}$ results in one of two cases:

1. If $|L_{\text{ratio}}| > c$, one of the reconstructions can be confidently determined as the more likely according to the parametric model. Then, the orientation is determined by the sign of $L_{\text{ratio}}$ and *CRISPR-evOr* returns "forward" or "reverse" accordingly.
2. If $|L_{\text{ratio}}| \leq c$, the likelihoods of the reconstructions are too close together and no confident prediction can be made. *CRISPR-evOr* returns "not determined".

A particularly common subgroup of uncertain orientation predictions is $|L_{\text{ratio}}| = 0$. Two such examples are shown in Fig B in S1 Material. In this case, the reconstructed events with forward orientation are exactly as likely as the reconstructed events with reverse orientation according to the parametric model. *SpacerPlacer* makes no difference between individual spacers (or arrays) but investigates only the retention of spacers and the number and location of deletion events [13]. Thus, for example a symmetric group of arrays (not considering spacer labels) will result in the same likelihoods and return $L_{\text{ratio}} = 0$. Notably, if the first and last spacer in each sample is identical this is obviously the case.

## Evaluation of acquisition-orientation

In Methods, we assess the performance of *CRISPR-evOr* in a simulation setting with three sets of simulated data generated as in [13]. Note that since every discussed orientation tool relies on distinct data types, it is not feasible to compare and test the performance of all discussed tools in a fair simulation setting.

In all simulated settings, *CRISPR-evOr* achieves a strong performance with an accuracy greater than 95% (with a threshold of $c = 0$). Most of the wrong predictions can be avoided, instead being labeled as uncertain, by setting a suitable threshold (e.g. $c = 5$). However, due to the limitations of the simulations and the relative power of the algorithm, the simulation setting cannot serve as a conservative estimator for a suitable choice of $c$ and thus requires evaluation on actual data.

In the CRISPRCasdb dataset, we found that many groups, e.g. the group shown in Fig 2, have high $|L_{\text{ratio}}|$ enabling *CRISPR-evOr* to make a clear decision for one orientation, when comparing both reconstructions. However, for array groups where limited or erratic evolution results in lower likelihood ratios, decisions can be more difficult and not always obvious (Fig C in S1 Material). Thus, we manually investigated the CRISPRCasdb data decisions with lower $|L_{\text{ratio}}|$ to determine a suitable conservative confidence threshold of $c = 5$, which indicates a difference of more than two orders of magnitude ($e^5$) between forward and reverse likelihoods. To further confirm the suitability of $c = 5$, we analyzed the difference in the number of deletion events between both reconstructions, as outlined in Methods. However, we recommend that users conduct their own experiments and manual investigations to ensure that the threshold is reliable for their particular use case and parameter settings.

We evaluate the performance of *CRISPR-evOr* by comparing our predictions to PAM-orientation predictions by Vink et al. [11] and the leader- and repeat-orientation methods CRISPRDirection [20] and CRISPRstrand [15] which are part of CRISPRCasFinder [19] and CRISPRidentify [16], respectively. The evaluation of PAM-orientation versus *CRISPR-evOr* and CRISPRDirection and CRISPRstrand versus *CRISPR-evOr* are performed on different datasets as reported in methods and in the respective figure captions.

Note that *CRISPR-evOr* considers groups of arrays, whereas the PAM-orientation and leader- and repeat-orientation methods consider single arrays. In all figures here, we show the orientation predictions for individual arrays. We show the results in terms of groups in Figs E and F in S1 Material. Moreover, we show pairwise comparisons between all tools in

Figs J-O in S1 Material. For array based methods, we arrive at group orientation predictions by following a consensus decision approach, see Note C in S1 Material. In Fig P in S1 Material we show a comparison of the fraction of confident predictions within each group for all three array-based tools.

**PAM- & acquisition-orientation are confident and agree for large parts of the dataset.** The number of arrays with confident predictions far exceeds those with uncertain predictions. However, both tools have a significant number of arrays (36.5%) where at least one of the tools is not confident in its predictions (Fig 3). As can be seen, PAM-orientation is confident on more arrays (89.3%) compared to *CRISPR-evOr* (68.6%). This is primarily due to many array groups that exhibit minimal or no evolution (113 groups with 471 arrays, 68.1% of all uncertain array predictions by *CRISPR-evOr*), making it challenging to predict the acquisition-orientation.

Additionally, we show in Fig 3 that *CRISPR-evOr* and PAM-orientation are almost always in agreement for abundant types I-E, I-B, I-F and II-A. With the only possible exception of type II-C, where PAM-orientation and *CRISPR-evOr* disagree in 21.3% of cases. However, the number of available groups and arrays of type II-C (178 arrays) is quite small. *CRISPR-evOr* is not confident in available type III-B groups as only one of them shows *any* evolution.

**Acquisition-orientation provides confident and reliable predictions compared to repeat based methods.** When we applied *CRISPR-evOr* on the CRISPRCasdb dataset, we found that our method is confident for 63.7% of the arrays (55.4% of the groups), whereas CRISPRDirection and CRISPRstrand are confident on 65.3% and 83.5% of the arrays, respectively. While, *CRISPR-evOr*, CRISPRDirection and CRISPRstrand are confident on the same samples for 35.4% of the arrays (Figs H and I in S1 Material), there is a substantial amount of samples (28.3%) where either CRISPRDirection and/or CRISPRstrand are uncertain but *CRISPR-evOr* is confident. Thus, *CRISPR-evOr* allows for confident orientation predictions for many arrays where other methods are unreliable.

Among the groups where predictions of *CRISPR-evOr* are not confident (231 of 518 groups with 2152 of 5934 arrays), the number of groups/arrays with $L_{\text{ratio}} = 0$ far exceeds the number of groups/arrays with some amount of evolution but no confident prediction (48 groups, 671 arrays), i.e. $0 < |L_{\text{ratio}}| \leq c$. Across the CRISPRCasdb dataset, 35.3% (183) of the groups, consisting of 25.0% (1481) of the arrays, have a likelihood ratio of 0 as they carry little to no diversity in spacer content or exhibit strong symmetry in forward and reverse reconstructions. Currently, on the subset of groups with sufficient non-symmetric diversity to use *CRISPR-evOr*, i.e. all 4453 arrays (335 groups) with $|L_{\text{ratio}}| > 0$, *CRISPR-evOr* is able to predict the orientation of 85.0% (3782) of the arrays (85.7% (287) of the groups) confidently.

To compare the contradictory predictions of the different tools, we performed a threefold comparison between *CRISPR-evOr* and CRISPRDirection and CRISPRstrand on the subset of our CRISPRCasdb dataset where *all* three tools have confident, but not necessarily identical, orientation predictions (Fig 4 and Figs F and G in S1 Material). This subset corresponds to the upper green column in Figs H and I in S1 Material, excluding rare types, amounting to 2071 arrays.

We find that for common well-studied Cas types like I-E and I-F, all tools are typically in agreement. For other systems, especially for type I-B and II-A, CRISPRstrand's and CRISPRDirection's prediction are more frequently in conflict than in agreement. This is remarkable since CRISPRDirection and CRISPRstrand rely mostly on similar information, i.e. the repeat mutation structure, although CRISPRDirection uses leader-based criteria as well. The independent evolutionary approach of *CRISPR-evOr* is able to resolve these conflicts. In general, our predictions are more aligned with CRISPRstrand's predictions than CRISPRDirection's predictions (Fig 4, Figs F and G in S1 Material).

The picture is different for Cas type II-C, where CRISPRDirection, and CRISPRstrand often disagree with each other, but *CRISPR-evOr* does not align consistently with either tool. Furthermore, *CRISPR-evOr* disagrees in many cases where CRISPRstrand and CRISPRDirection are in agreement.

## Discussion

Here, we demonstrate that the polarized insertion of spacers enables inference of the orientation of CRISPR arrays using evolutionary likelihood calculations. Compared to sequence-based methods, this is an independent approach, which not

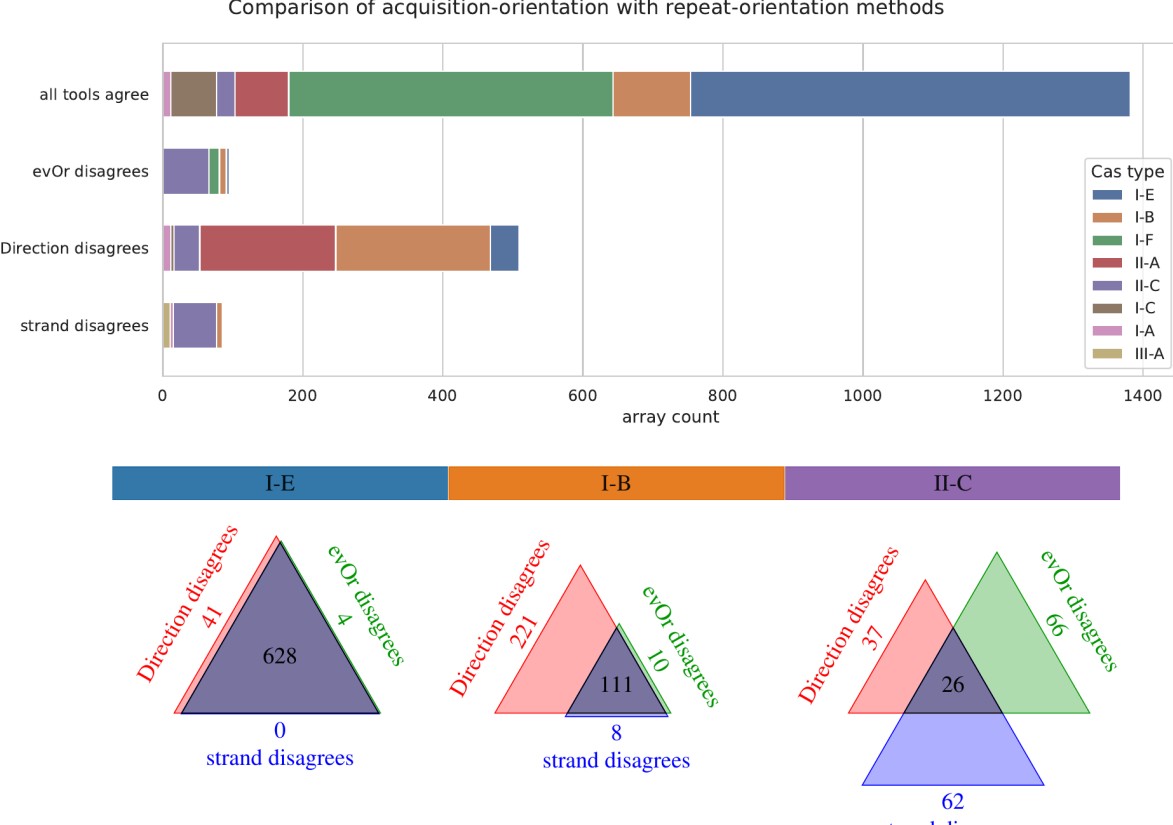

**Fig 4**. **Comparison with repeat based orientation methods.** We show the agreement of orientation predictions between CRISPRDirection, CRISPRstrand and *CRISPR-evOr* where all methods have high confidence broken down according to Cas type. Cas types were filtered as in Fig 3. At the bottom, we show Venn diagrams of the same data for three interesting subtypes with distinct behavior. Clearly, all methods agree quite often, especially for well-researched Cas types like I-E and I-F. However, there are substantial differences for predictions for type I-B, II-A and II-C. This indicates differences in the characteristics used for orientation prediction such as their repeats, leader and acquisition behavior. CRISPRDirection and CRISPRstrand were developed/trained using the more prevalent and well-studied Cas types, they may struggle for specific less common Cas types. Venn diagramms for the remaining types can be found in Fig G in S1 Material.

only allows to predict the orientation of previously unpredictable arrays with high confidence, but also to reject or confirm previous sequence-based CRISPR array orientation predictions.

## Challenges in accurately predicting the array orientation

An advantage of our evolutionary approach in comparison with other approaches is that the detection of leader, repeat, and PAM sequences, which can be unreliable due to their inherent variability, does not influence our predictions. For example, some CRISPR-Cas systems do not have (a clearly identifiable) leader [9]. Orientation decisions based on repeats are often reliable, but also often lack confidence, particularly for short CRISPR arrays with few repeats [16,19].

Similarly, the transcription orientation of the Cas genes and the array orientation frequently but not necessarily align. Moreover, not for all CRISPR arrays Cas genes can be found nearby, and if so, CRISPR-Cas systems (e.g. I-A, II-C2) can also contain Cas genes that are transcribed in different directions [4].

For PAM-based orientation predictions, efficiently and reliably identifying protospacers and PAMs remains a challenge and depends on the availability of appropriate metagenomic datasets. Nevertheless, in those cases where this is possible,

we consider the orientation inferred from PAMs to be a reliable technique for determining the array direction, particularly when the results are based on a multitude of spacers within an array. However, one has to consider that the location and properties of PAM sequences are subtype specific. For example, the PAM is known to be on the 5' end for type I, type IV and V, and on the 3' end for type II systems where the PAM is also strain- and not repeat-specific, and not all types have (known) PAMs [11,33,34].

### An evolutionary approach for array orientation prediction

The acquisition-orientation considered here is a distinct concept independent of (but correlated with) the leader- and repeat-orientation concepts. Unlike all other tools discussed, *CRISPR-evOr* requires multiple CRISPR arrays with some amount of overlap and differences.

Naturally any evolutionary approach can not provide an orientation estimate for a single array or a set of identical arrays. In CRISPRCasdb, we found a substantial amount of closely related arrays which do not show observable evolution. We believe that besides CRISPR systems that lost their acquisition machinery, global sampling biases towards more closely related strains contribute to the current lack of diversity among these spacer arrays.

Ongoing sequencing and sampling efforts will extend the available datasets and generate larger more diverse groups of related arrays, and, in turn, allow *CRISPR-evOr* to make even more confident and reliable predictions in the future when additional arrays become available. However, *CRISPR-evOr* is already able to make confident decisions based on groups with only three arrays and sufficient diversity.

Our approach to predict orientation based on the diversity of CRISPR arrays is not limited to the concrete likelihood calculation based on the evolutionary block deletion model in *SpacerPlacer*. In principle, any evolutionary model, such as the most parsimony approach implemented in the CRISPR Comparison Toolkit (CCTK) [35] could be used to reconstruct ancestral arrays for both orientations and identify the more parsimonious or likely direction.

Variations of the acquisition and deletion mechanisms between different types, which differ from model assumptions, could impact any evolutionary approach. While the deletion mechanisms are rather universal across different subtypes [13], some Cas subtypes do not always strictly follow polarized insertions. However, polarized insertions appear to be the predominant mode of spacer acquisition [13], although in some studies ectopic insertions within the array have been observed for I-A [36,37] and II-A [38–41]. However, in all observed cases they occur less frequently than polarized insertions and only when specific regions of the leader are mutated or lost [36–38]. *CRISPR-evOr* performs well on available II-A arrays which supports results showing that type II-A acquisition is typically polarized [38].

Indeed, *CRISPR-evOr* is more decisive on simulated data, based on a simple evolutionary model, than real data, suggesting that CRISPR arrays are subject to more complex evolutionary processes. This not only includes uncommon events like ectopic insertions, duplicate events, and array collapse, but also sequencing errors, selection pressures, and changes in the environment which can all lead to unexpected patterns. However, spacer duplications and other complex evolutionary events are classified as such by *SpacerPlacer* and thus do not directly impair the orientation prediction of *CRISPR-evOr*.

In conclusion, although there are various issues that can impede accurate prediction of acquisition orientation, there are three essential requirements for reliable predictions by *CRISPR-evOr* that can be summarized as: (i) a sufficiently diverse set of different, but related, CRISPR arrays with (ii) active evolution and (iii) polarized acquisition of new spacers. The strong performance in simulations in combination with the agreement with sequence based prediction methods for most CRISPR system types underscores the power of our evolutionary approach.

### Comparison of orientation predictions across Cas types

We find that the orientation predictions of the abundant Cas types I-E and I-F show a strong consensus among all investigated orientation concepts. The consensus is to be expected as these abundant types are well-researched and served

as blueprint for typical CRISPR behavior in terms of evolution, expression, transcription and repeat and leader characterization. CRISPRDirection, CRISPRstrand and CRISPRleader rely on leader and repeat attributes that were defined based on these most abundant types, which are also a substantial part of the training and test datasets for CRISPRstrand and CRISPRdirection. Similarly, *CRISPR-evOr* is reliant on polarized insertion, which is a behavior that characterizes (almost) all types of CRISPR arrays but was identified and experimentally confirmed particularly for type I.

Interestingly, for types I-B and II-A, *CRISPR-evOr*, PAM-orientation and CRISPRstrand agree in their prediction for almost all arrays, while CRISPRDirection disagrees with them. Thus, for those types, CRISPRstrand is likely more reliable than CRISPRDirection. Since the orientation predictions are reinforced by three different concepts based on the PAM, the repeats and the evolutionary likelihood, we conclude that CRISPRDirection is likely misclassifying the corresponding arrays. This is in line with the observation that CRISPRDirection is also confident for fewer arrays than CRISPRstrand (Figs H and I in S1 Material).

While wrongly predicted orientations by CRISPRDirection and, consequently, in CRISPRCasdb have already been noted for some strains and systems [15,42], our results indicate that—at least among the considered dataset of arrays, with confident predictions by CRISPRDirection, CRISPRstrand and *CRISPR-evOr*—the orientations of 63.1% of type I-B and 71.6% of II-A arrays on CRISPRCasdb are incorrect.

Correctly inferred orientations are crucial for a multitude of subsequent analyses, e.g. regarding array transcription, or protospacer identification [43–46]. Thus, for the afore mentioned and related CRISPR types we advise reassessing the corresponding data and experiments to verify if the conclusions remain valid when the orientation of many arrays has to be reversed.

Among the three tools, the greatest divergence in predicting the array orientation arises with type II-C (Figs 3 and 4, Fig G in S1 Material). Type II-C has been shown to be particularly challenging to orient due to its unusual features in spacer acquisition, expression, leader, repeat structure [22,24–26], and PAM diversity [11]. In particular, multiple studies reported that II-C arrays favor acquisition at the downstream 3'-end of the array [22–24].

Moreover, type II-C has not only leader promoters, but promoters in the repeats [22,25], which lowers the selective pressure to insert new spacers at the leader end and may cause deviations from patterns typically observed in leaders and repeats of other CRISPR systems.

However, as long as the (majority of) acquisitions are polarized and the acquisition end is consistent *within individual groups*, the predictions of *CRISPR-evOr* likely remain accurate and consistent. Indeed, manual inspection of the reconstructions of the groups of II-C arrays with confident predictions showed normal (polarized) evolutionary behavior and thus sensible acquisition-orientation predictions.

Thus, the inconsistency of acquisition-orientation with both methods to predict repeat-orientation suggests that type II-C arrays might be composed of subgroups with different acquisition ends and thus inconsistency within (leader and) repeat structure. This could cause the complex discrepancies between acquisition-orientation and repeat-orientation.

Multiple subtypes of II-C have been proposed in [4] based on the composition of their Cas loci and recent results in [47] point into the same direction, where we found that repeat diversity clusters II-C in multiple groups and could be related to co-occurrence of other Cas types. In any case, the inability of the tools to determine consistent orientations for type II-C warrants further investigation of the diversity and functionality of II-C systems.

## Conclusions and future perspectives

With *CRISPR-evOr* we provide a reliable tool to predict the acquisition-orientation of CRISPR arrays. Our results suggest to treat the orientations of CRISPR arrays from CRISPRCasdb, at least for some systems, with caution.

Among the different orientation concepts, acquisition-orientation is most directly related to the adaptive evolutionary dynamics of spacer arrays and thus carries an additional ecological importance. The acquisition-orientation is crucial to

understand the interaction between the environment and the CRISPR-Cas system. This includes especially the acquisition and deletion mechanisms, and their defensive and/or regulatory function in bacteria and archaea. In particular, to assess the relevance and protective capabilities of specific spacers and the ecological importance of their targets, it is essential to know whether spacers have been acquired recently or in the more distant past [48].

As the availability of CRISPR arrays continues to grow, especially with the increasing number of long-read sequencing projects, *CRISPR-evOr* will allow to investigate CRISPR arrays for various CRISPR-Cas types in the future, including type III CRISPR arrays that have been found to be distinct in their spacer acquisition [39,45].

## Supporting information

**S1 Table. CRISPRCasdb dataset groups with orientation predictions, accession numbers and other useful information.** The column headers are described in Note E in S1 Material.
(CSV)

**S1 Material. Includes Supplemental Notes A–E providing additional details and remarks, and Supplemental Figs A–P.**
(PDF)

## Acknowledgments

We thank Jochem N. A. Vink, Stan J. J. Brouns for valuable comments and discussions and providing the updated PAM-orientation dataset. We also thank Peter C. Fineran for helpful comments and discussions.

## Author contributions

**Conceptualization:** Axel Fehrenbach, Franz Baumdicker.

**Data curation:** Axel Fehrenbach, Alexander Mitrofanov, Omer S. Alkhnbashi.

**Formal analysis:** Axel Fehrenbach, Franz Baumdicker.

**Funding acquisition:** Rolf Backofen, Franz Baumdicker.

**Investigation:** Axel Fehrenbach, Alexander Mitrofanov.

**Methodology:** Axel Fehrenbach, Franz Baumdicker.

**Software:** Axel Fehrenbach, Alexander Mitrofanov, Omer S. Alkhnbashi.

**Supervision:** Rolf Backofen, Franz Baumdicker.

**Visualization:** Axel Fehrenbach, Franz Baumdicker.

**Writing – original draft:** Axel Fehrenbach.

**Writing – review & editing:** Axel Fehrenbach, Alexander Mitrofanov, Omer S. Alkhnbashi, Rolf Backofen, Franz Baumdicker.

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
