## [Decision Letter · Decision Letter 0]

30 Jul 2025

PCOMPBIOL-D-25-00939

An evolutionary approach to predict the orientation of CRISPR arrays

PLOS Computational Biology

Dear Dr. Fehrenbach,

Thank you for submitting your manuscript to PLOS Computational Biology. After careful consideration, we feel that it has merit but does not fully meet PLOS Computational Biology's publication criteria as it currently stands. Therefore, we invite you to submit a revised version of the manuscript that addresses the points raised during the review process.

Please submit your revised manuscript within 30 days Sep 29 2025 11:59PM. If you will need more time than this to complete your revisions, please reply to this message or contact the journal office at ploscompbiol@plos.org. Please include the following items when submitting your revised manuscript:

We look forward to receiving your revised manuscript.

Kind regards,

Lingchong You

Academic Editor

PLOS Computational Biology

Natalia Komarova

Section Editor

PLOS Computational Biology

**Additional Editor Comments :**

Please consider all the comments provided by the referees. Specifically it has been suggested that the presentation and logic of the manuscript should be improved.

**Journal Requirements:**

At this stage, the following Authors/Authors require contributions: Axel Fehrenbach, Alexander Mitrofanov, Omer Alkhnbashi, Rolf Backofen, and Franz Baumdicker. Please ensure that the full contributions of each author are acknowledged in the "Add/Edit/Remove Authors" section of our submission form.

4) Your manuscript is missing the following section heading: Abstract.  Please ensure all required sections are present and in the correct order. Make sure section heading levels are clearly indicated in the manuscript text, and limit sub-sections to 3 heading levels. An outline of the required sections can be consulted in our submission guidelines here:

5) Please upload all main figures as separate Figure files in .tif or .eps format. For more information about how to convert and format your figure files please see our guidelines: 

6) We have noticed that you have uploaded Supporting Information files, but you have not included a list of legends. Please add a full list of legends for your Supporting Information files after the references list.

7) Please amend your detailed Financial Disclosure statement. This is published with the article. It must therefore be completed in full sentences and contain the exact wording you wish to be published.

1) State the initials, alongside each funding source, of each author to receive each grant. For example: "This work was supported by the National Institutes of Health (####### to AM; ###### to CJ) and the National Science Foundation (###### to AM)." Please include the initials of the author who received these grants "BA 2168/23-1, BA 2168/23-2, and the grants provided by the German Network for Bioinformatics Infrastructure.

3) If any authors received a salary from any of your funders, please state which authors and which funders.

8) Please ensure that the funders and grant numbers match between the Financial Disclosure field and the Funding Information tab in your submission form. Note that the funders must be provided in the same order in both places as well. If you receive funds from " the Multiple Functions and Facets of CRISPR–Cas; Baden-Wuerttemberg Ministry of Science, Research and Art; University of Freiburg," please include them in the Funding Information tab.

**Reviewers' comments:**

Reviewer's Responses to Questions

Reviewer #1: This study addresses the problem of determining the correct orientation of CRISPR arrays, which is essential for understanding their function and reconstructing their evolutionary history. Existing tools often rely on features such as leader sequences, PAMs, Cas genes, or repeat mutations, which can be inconsistent or absent. The authors present CRISPR-evOr, a new algorithm that infers orientation by reconstructing the evolutionary history of closely related CRISPR arrays in both directions. By comparing their likelihoods, they determine which orientation is more likely.

This work/algorithm is built upon their previous work: Fehrenbach, Axel, et al. "SpacerPlacer: ancestral reconstruction of CRISPR arrays reveals the evolutionary dynamics of spacer deletions." Nucleic Acids Research 52.18 (2024). It’s implemented as a function of SpacerPlacer. Essentially, this work used SpacerPlacer’s concept but applied to a unique problem.

The concept is interesting. The core novelty is calculating both forward and reverse scenarios and using a log-likelihood ratio to assess which direction is more plausible. One of the technical concepts is how to define a confidence threshold for making reliable orientation predictions.

While I found the conceptual framework interesting, the manuscript is poorly written and structured. I found it very confusing to read and grasp the idea, and the figures need to be better designed to demonstrate the concepts. Also some of the key ideas need to be explained. Please see my specific comments.

Major

1. Introduction, lines 133–134, right column, on the “low-confidence” areas, the statement “Thus, our tool increases the number of arrays where orientations can be reliably estimated” should be more informative with a quantitative performance comparison. This is a main problem with the maintext, it’s more descriptive than quantitative.

2. In Methods, the authors said “These arrays were grouped successively by repeat, genus, Cas type to yield a large comprehensive dataset.” “Group” is a critical concept for the evolution tree construction. But it is not sufficient explained, and too abstract to evaluate the algorithm built on this. Including an example could be helpful. Perhaps as part of Figure 2.

3. The left panels of Figure 1A, B could use a larger font size. Besides, I found this figure less essential to the main concepts and suggest removing it or move to supplementary material, and use the current figure 2 as figure 1, as it captures the concept of the algorithm. In figure 2, the author should label which step is done by which part of the algorithm.

4. The authors mentioned using SpacerPlacer in the introduction but left it unexplained. It is also not mentioned that this comes from the same authors. Explaining the previous work is important, it will help the readers understand the novelty of this work and how it’s tied to the rest of the field and the authors’ research goal.

5. The Result section is a mixture of methods+results+discussion. Please condense and move some of the content to other sections. “Orientation algorithm”, “Description of orientation algorithm”, “Likelihood comparison and measuring confidence” should be condensed into one subsection, i.e. the algorithm. Part of “Simulation-based benchmarking.” And “Evaluation of acquisition-orientation on real data” should be moved to methods, important results can be kept.

6. The work used several datasets, such as CRISPRCasdb and simulated, and benchmarked CRISPR-evOr against existing tools showing good performance in both simulated and real datasets. Please help the readers how do they look like and your data diversity. could use a figure.

7. The threshold c is chosen empirically, based on simulation and manual inspection. While this approach is understandable for exploratory work, I believe a more formal justification is necessary, such as hypothesis testing, statistical modeling, or heuristic optimization. Please also discuss how this problem has been solved by the broader research community, not necessarily on this topic, but evolution, longitudinal data in general.

Minor

1. Figure 3b, bottom stack, around 600 array count tick, the purple and blue bars are only PAM not confident. But the entire bottom stack should also be crispr-evor not confident?

2. Introduction, the comparison between “low confidence,” “high confidence,” and “available orientation predictions” is unclear. The distinction between low and high confidence makes sense and directly relates to method performance: CRISPR-evOr helps in low-confidence cases and tends to agree with other tools when high-confidence predictions are available. However, the “available orientation” category is conceptually distinct. If this refers to annotated orientations, which are used as ground truth or priors for the algorithm, this should be clearly stated and justified in the text. But based on the algorithm description, I don’t think this is the case.

3. The discussion is overly long, please make it shorter

The code is available as a function within the SpacerPlacer package on GitHub, with no standalone script provided.

Reviewer #2: This is a nice manuscript in which the authors develop and test a new tool, CRISPR-evOr, for detection of the direction of CRISPR arrays based on evolutionary history of spacer acquisition patterns. This can only works for arrays where several related CRISPR arrays are available, but in such cases the method can resolve low confidence or contradictory predictions from existing tools, which is very useful. My specific comments follow below.

“Moreover, CRISPR-evOr, 205 PAM-orientation, CRISPRDirection, and CRISPRstrand often disagree with each other for type II-C, which hints at differences between acquisition-orientation and leader-, repeat- and PAM-orientation for type II-C” – it is fairly well established that II-C systems often have a different spacer acquisition orientation than other systems which is at the leader distal rather than the leader proximal end. This should be discussed, since it “derails” other tools also, but affects CRISPR-evOr the most (see for example the last paragraph of the Results. The authors only get to it in the discussion, where their interpretation is not clear enough for the reader to follow.

372 “ reconstruction process we identify and relabels unique” – fix

485 – fix ref (add number)

**Have the authors made all data and (if applicable) computational code underlying the findings in their manuscript fully available?**

Reviewer #1: Yes

Reviewer #2: Yes

PLOS authors have the option to publish the peer review history of their article (what does this mean?). If published, this will include your full peer review and any attached files.

Reviewer #1: No

Reviewer #2: No

**Figure resubmission:**
---

## [Decision Letter · Decision Letter 1]

4 Nov 2025

Dear Mr Fehrenbach,

We are pleased to inform you that your manuscript 'An evolutionary approach to predict the orientation of CRISPR arrays' has been provisionally accepted for publication in PLOS Computational Biology.

Best regards,

Lingchong You

Academic Editor

PLOS Computational Biology

Natalia Komarova

Section Editor

PLOS Computational Biology

Reviewer's Responses to Questions

**Comments to the Authors:**

Reviewer #1: The authors have addressed my comments.

Reviewer #2: All my comments have been addressed in the revised version

**Have the authors made all data and (if applicable) computational code underlying the findings in their manuscript fully available?**

Reviewer #1: Yes

Reviewer #2: Yes

PLOS authors have the option to publish the peer review history of their article (what does this mean?). If published, this will include your full peer review and any attached files.

Reviewer #1: No

Reviewer #2: No

---

## [Editor Report · Acceptance letter]

PCOMPBIOL-D-25-00939R1

An evolutionary approach to predict the orientation of CRISPR arrays

Dear Dr Fehrenbach,

I am pleased to inform you that your manuscript has been formally accepted for publication in PLOS Computational Biology. Your manuscript is now with our production department and you will be notified of the publication date in due course.

With kind regards,

Anita Estes
